# Protective Effect of Flavonoids against Methylglyoxal-Induced Oxidative Stress in PC-12 Neuroblastoma Cells and Its Structure–Activity Relationships

**DOI:** 10.3390/molecules27227804

**Published:** 2022-11-12

**Authors:** Danyang Zhang, Xia Li, Xiaoshi He, Yan Xing, Bo Jiang, Zhilong Xiu, Yongming Bao, Yuesheng Dong

**Affiliations:** 1School of Bioengineering, Dalian University of Technology, Dalian 116024, China; 2School of Ocean Science and Technology, Dalian University of Technology, Panjin 124221, China

**Keywords:** flavonoids, methylglyoxal, PC-12 cells, structure–activity relationship, oxidative stress, molecular docking

## Abstract

Methylglyoxal-induced oxidative stress and cytotoxicity are the main factors causing neuronal death-related, diabetically induced memory impairment. Antioxidant and anti-apoptotic therapy are potential intervention strategies. In this study, 25 flavonoids with different substructures were assayed for protecting PC-12 cells from methylglyoxal-induced damage. A structure–activity relationship (SAR) analysis indicated that the absence of the double bond at C-2 and C-3, substitutions of the gallate group at the 3 position, the pyrogallol group at the B-ring, and the *R* configuration of the 3 position enhanced the protection of flavan-3-ols, and a hydroxyl substitution at the 4′ and meta-positions were important for the protection of flavonol. These SARs were further confirmed by molecular docking using the active site of the Keap1–Nrf2 complex as the receptor. The mechanistic study demonstrated that EGCG with the lowest EC_50_ protected the PC-12 cells from methylglyoxal-induced damage by reducing oxidative stress via the Nrf2/Keap1/HO-1 and Bcl-2/Bax signaling pathways. These results suggested that flavan-3-ols might be a potential dietary supplement for protection against diabetic encephalopathy.

## 1. Introduction

Diabetic encephalopathy, a serious diabetic complication of a microvascular nature, is chiefly a consequence of central nervous system disease [1]. It is characterized by mild cognitive decline, which can progress to dementia in severe cases [2]. As reported in the literature, people with diabetes have a 60% higher risk of developing dementia [3]. Moreover, diabetic encephalopathy is tightly associated with long-term hyperglycemia in diabetic patients, owing to oxidative stress [4], inflammation [5], and an abnormal accumulation of advanced glycation end products (AGEs) [6]. Methylglyoxal (MG) is recognized as the most likely AGE precursor contributing to intracellular AGE formation [7].

In vivo, MG is derived from glucose metabolism, amino acid metabolism, fat metabolism, and, most importantly, the glycolysis pathway [8], with a low level under normal circumstances. However, an excessive MG level can be cytotoxic, leading to irreversible cell death [9] owing to an increase in oxidative stress [10]. As we know, it has already been reported that the concentration of MG is abnormally elevated in diabetics [11]. It is also implied that an over accumulation of MG caused by abnormal glucose metabolism is a major cause of diabetic encephalopathy [12]. Therefore, the intervention of oxidative stress is considered an effective strategy for preventing diabetic encephalopathy. Nuclear factor erythroid 2-related factor2 (Nrf2) is described as a master redox-related transcription factor, maintaining intracellular redox homeostasis [13]. When oxidative stress occurs, the binding of Nrf2 and Keap1 becomes unstable, and Nrf2 is released, transferring from the cytoplasm to the nucleus. This, in turn, can activate the expression of downstream antioxidant enzymes, including HO-1, a kind of phase II enzyme [14]. It was verified that the expression of Nrf2 decreased and that of Keap1 increased in HLECs after a treatment of MG [9]. As a result, the damage induced by MG can be alleviated by interventions in the Nrf2/Keap1 pathway.

Recently, it has been accepted that dietary supplementation has become a noteworthy intervention for improving glucose metabolism disorders and cognitive impairment. Flavonoids, a group of phenolic compounds, can be found in many fruits and vegetables, such as apples, scutellaria, celery, and so on [15], which are composed of two aromatic rings (A and B) connected to each other through a central three-carbon chain, generally with C6–C3–C6 as the basic skeleton. Increasing evidence has highlighted that flavonoids have abundant properties, including those anti-diabetic [15], anti-dementia [16], anti-tumor [17], anti-osteoporosis [18], anti-arteriosclerosis [19], antioxidant [15], and so on. The SAR of flavonoids has been studied in a cellular antioxidant activity assay using HepG2 cells [20]. The study clarified that a 3′,4′-O-dihydroxyl group in the B-ring, a 2,3-double bond combined with a 4-keto group in the C-ring, and a 3-hydroxyl group enhanced antioxidant capacity to treat cancer. It was also reported that the SAR of grape seed procyanidins against H_2_O_2_-induced oxidative stress in PC-12 cells could be used to treat neurodegenerative disorders [21]. The authors claimed that there was a positive correlation between the procyanidins’ polymerization degree and the protective effect against oxidative stress in PC-12 cells and that the presence of 3-galloylated groups increased the protective activity. However, procyanidins are a kind of flavan-3-ols polymer and the research on the SAR of the flavna-3-ols monomer and other flavonoids in diabetic encephalopathy is not comprehensive.

To the best of our knowledge, a systematic evaluation of the protection offered by flavonoids with different substructures against damage induced by MG in nerve cells has not been reported, and the SAR between the flavonoids and the protective activities against the toxicity of MG in neurons is also scarcely documented. Consequently, the design of dietary supplements for protection against diabetic encephalopathy is urgently required. It was indicated that hesperitin resisted oxidative stress via ER- and TrkA-mediated actions in PC-12 cells [22]. Prior research also showed that dihydromyricetin ameliorated oxidative stress via the AMPK/GLUT4 signaling pathway in MG-induced PC-12 cells [23].

In this study, the protective effects of different kinds of flavonoids against the damage induced by MG in PC-12 cells were determined. The SARs of these flavonoids were also analyzed and summarized. Furthermore, to understand the mechanism, the effects of the flavonoids with the highest activity on the signaling pathway related to oxidative stress were also investigated in the PC-12 cells.

## 2. Results

### 2.1. Cytotoxicity of Flavonoids to PC-12 Cells

In order to eliminate the effect of cytotoxicity in the activity assay of the flavonoids, the cytotoxicity test was initially performed using an MTT assay. The results showed that the flavonoid concentrations below 100 μM exhibited no cytotoxicity to PC-12 cells. The consequent protective activity was assayed and the concentration ranged below 100 μM for each flavonoid without apparent cytotoxicity to the PC-12 cells.

### 2.2. The Protective Activities and SAR Analysis of Different Flavonoids With Respect to the Damage Induced by MG

Concentrations from 0.25 to 4 mM of MG were added to PC-12 cells after they had incubated for 48 h and dose-responses of MG were detected by the MTT assay concerning the effect of MG on the damage to the PC-12 cells. Finally, the IC_50_ value of MG was selected as 0.5 mM after 48 h (Appendix A), which was chosen based on the PC-12 cells in order to examine the protective effect of the flavonoids in the following study. Curcumin was selected as the positive control (EC_50_ = 1.31 ± 0.42 μM) (Appendix A). A total of 25 flavonoids consisting of five types—5 flavan-3-ols, 5 flavonols, 2 flavanones, 8 flavones, and 5 isoflavones (Figure 1 and Appendix A)—were assayed to evaluate their protective effects on the cell viability of the PC-12 cells stimulated by MG using the MTT method. The activities were expressed as EC_50_ [24], and the data were summarized in Table 1. The compounds whose EC_50_ values were less than 100 μM were regarded as active compounds.

The results indicated that the skeleton structure of the flavonoids had a crucial impact on their protective activities. Generally, most of the compounds in the flavan-3-ol subtypes exhibited the strongest protective activities, followed by the flavanone subtype, and the protective activity of most compounds under the flavonol subtype were weaker than the former two subtypes. In contrast, neither the compounds in the flavonoid subtype nor the isoflavone subtype showed protective activity.

For the subtype of the flavan-3-ols, EGCG and ECG showed stronger protective activities than those of EGC and EC (11.98 ± 0.49 μM vs. 43.06 ± 1.18 μM, and 34.52 ± 2.69 μM vs. 52.34 ± 3.99 μM), suggesting that the gallate group substitution enhanced the protective activity of the MG-treated PC-12 cells. Moreover, it was also crucial for the presence of the pyrogallol substitution on B-ring, since the protective effect of EC was weaker than EGC (52.34 ± 3.99 μM vs. 43.06 ± 1.18 μM) on the MG-induced PC-12 cells. Interestingly, it was found that the configuration of the 3 position in the flavan-3-ols played an important role in their protective activity. All the substitutions at the 3 position with an R configuration (EC, EGC, ECG, and EGCG) showed relatively strong protective activities; that is, no obvious protective activity was found for the (+) catechin, whose substitution at the 3 position was in the S configuration. A previous study also indicated that flavan-3-ols with the galloyl moiety substation and an R configuration at the 3 position had improved antioxidant activity in HepG2 tumor cells [20], suggesting that the substitution of the galloyl moiety and the R configuration at the 3 position are important for the protective activity of flavan-3-ols in different cell lines.

Regarding the flavanone subtype, both tested compounds showed protective activities. In addition, the naringenin showed relatively strong activities (EC_50_ = 13.35 ± 1.92 μM), whereas no obvious activity was observed in apigenin, whose structure is almost identical with naringenin, wherein the only difference is the existence of a double bond between C-2 and C-3. These data were consistent with the result obtained for the flavan-3-ols subtype wherein the absence of a double bond at the C-2 and C-3 position was important to their protective activities. However, it was reported that naringenin did not show obvious activity in a cellular antioxidant activity (CAA) assay of HepG2 cells [20]. These data indicated that the influence of the double bond between C-2 and C-3 in flavanone on its cell protective activity is cell-line-specific. Of course, as the commercially available flavanone subtypes are limited, only two of them were tested for their protective activities. In the future, other SARs—besides the importance of the bond type between the C-2 and C-3—need to be summarized after the assaying of more flavanones with respect to their protective activities in the MG-treated PC-12 cells.

For the flavonol subtype, quercetin showed higher activities (EC_50_ = 28.83 ± 2.76 μM) than luteolin (EC_50_ > 100 μM). Meanwhile, the substitution of the hydroxyl group at the 3 position with a glycosyl group weakened flavonol’s protective activities; for example, the substitutions of the hydroxyl group at the 3 position in quercetin with either glucosidase (isoquercitrin) or rutinosidase (rutin) nullified flavonol’s protective activities. These results demonstrated the importance of the hydroxyl group substitution at the 3 position on the C-ring. The hydroxyl group substitution on the other position also influenced its protective activities: higher activities of quercetin (EC_50_ = 28.83 ± 2.76 μM) than kaempferol (EC_50_ > 100 μM) indicated that the 4′-hydroxyl group played an important part when the 3′-hydroxyl group was present. Moreover, it was clarified that meta-hydroxyl groups are stronger than the ortho-hydroxyl groups on the B-ring by comparing morin (EC_50_ = 14.83 ± 1.70 μM) to quercetin (EC_50_ = 28.83 ± 2.76 μM). However, quercetin showed stronger activity than morin in the CAA assay [20]; thus, its mechanism needs to be further explored.

### 2.3. Molecular Docking Study

To further explain the SARs of the flavonoids, in silico docking studies were performed using Autodock software. Consequently, it was suggested that the Keap1–Nrf2 complex was the target protein that the drug interacted with in the antioxidative signaling pathway [25]; therefore, the Keap1–Nrf2 complex (PDB ID: 2FLU [25]) was selected as the receptor in the docking study. For the subtype of the flavan-3-ols, the binding energy and the hydrogen bonds of the major flavonoids with the complex were calculated and summarized in Table 1. For the binding energy, EGCG showed the lowest binding energy (−40.041 kJ/mol), followed by ECG, EGC, and EC, and (+)-Catechin showed the highest binding energy. The protective activities and binding energy showed the same tendency, which supported the importance of the gallate and pyrogallol groups’ substitution as well as the 3 position in the flavan-3-ols found in the SAR analysis. The results of the SAR analysis in the assay could also be confirmed by hydrogen bond data (Table 1). In the best conformation of EGCG (Figure 2A), a gallic acid substituent formed five of the eight hydrogen bonds (H-bond) with three amino acids in the binding pocket (Val 561, Arg 326, and Val 369), and the hydroxyl groups on the basic skeleton formed the rest of the three hydrogen bonds with the other three amino acids in the binding pocket (Val 467, Val 418, and Val 420). Similarly, compared with EGCG, EGC, whose structure lacked a gallic acid substituent, was predicted to only form four H-bonds between the hydroxyl groups in the 3 position and B ring and with Val 418, Leu 557, and Val 604, which demonstrated that the presence of a gallic acid substituent could increase the activities in question (Figure 2A). The actual EC_50_ and predicted binding energy are presented in Appendix A, and a good linear correlation (R^2^ = 0.7084) was observed. The binding energy reflected the neuroprotective activity against MG-induced damage and provided information for further drug screening.

For the flavanone subtype, the binding energies of naringenin and apigenin were calculated as −35.271 kJ/mol and −33.639 kJ/mol, respectively, which also showed the same tendency with the assay. The H-bond data indicated that both the two compounds formed three H-bonds with the amino acids. Naringenin formed H-bonds with Gly 367, Val 606, and Leu 557 (Figure 2B), while apigenin formed H-bonds with Val 514, Gly 367, and Val 418(Figure 2B). It was supposed that a single bond in naringenin could be twisted, thus alternating the docked position, which affected the binding energy.

For the flavonol subtype, quercetin showed a relatively lower binding energy of −34.351 kJ/mol, forming four H-bonds between hydroxyl groups and amino acid residue; for instance, the 3′-hydroxyl group formed two hydrogen bonds with Val 465 and Val 512, the 3-hydroxyl group formed one hydrogen bond with Val 606, and another hydrogen bond formed between the 4′- hydroxyl group and Val 418 (Figure 2C). On the contrary, both luteolin (Figure 2C) and kaempferol (Figure 2D), which lack the 3-hydroxyl and 3′- hydroxyl groups, respectively, only formed three H-bonds and, consequently, showed higher binding energy. Moreover, it was calculated that morin had a lower binding energy than quercetin, namely, −34.895 kJ/mol, which might be because morin formed two key H-bonds between the 2′- hydroxyl group with Val 606 and Gly 367 (Figure 2E). On the other hand, compared with quercetin, the docking data indicated that the binding energies were higher in rutin (Figure 2F) and isoquercitrin, which might be due to the glycoside group substitution that blocked the entry of the compounds into the binding pocket. All these docking results were consistent with the assay data, and further confirmed the importance of the substitution of hydroxyl groups in the 2′, 3′, and 3 positions in the flavonol subtype.

In this study, the use of MG-induced PC-12 cells through an in vitro method were established to test the neuroprotective activity of the flavonoids, and a silicon molecular-docking method using the Keap1–Nrf2 complex as the receptor was used to further verify the SAR analysis. The in vitro assay and the silicon showed almost the same tendency. These results suggested that the docking of the test compounds with the Keap1–Nrf2 complex prior to the in vitro assay might constitute an alternative method for the neuroprotective assay, which could lower costs and greatly increase efficiency. On the other hand, only 25 flavonoids were tested in our study; thus, more flavonoids need to be tested to verify the results of the SAR analysis.

### 2.4. The Mechanism of EGCG on MG-Induced Oxidative Stress and Apoptosis in PC-12 Neuroblastoma Cells

Previous studies and our docking experiments demonstrated that the Nrf2-Keap1 complex and antioxidants play an important role in the protective activities of the flavonoids against the nerve cell damage induced by MG. Thus, the mechanism of the flavonoids against MG-induced oxidative stress in PC-12 neuroblastoma cells was studied through a Western blot analysis of the Nrf2/Keap1/HO-1 signaling pathway. As it was shown, after the MG treatment, the expression of HO-1 decreased, while the expression of Keap1 noticeably increased. However, EGCG significantly promoted the expression of HO-1 and inhibited the expression of Keap1 in a dose-dependent manner (Figure 3A–D). Meanwhile, the results showed that EGCG induced the nuclear translocation of Nrf2 (Figure 3E,F). This suggested that the intervention of the Nrf2/Keap1/HO-1 pathway, which is closely related to oxidative stress, is one of the key mechanisms of EGCG-mediated neuroprotection against MG-induced PC-12 cells.

Oxidative stress can cause neuronal apoptosis [26]. To investigate the effect of EGCG on the MG-induced PC-12 cells’ apoptosis, PC-12 cells were co-cultured with EGCG in an MG-containing medium. The flow cytometry results showed that MG-induced PC-12 cells’ apoptosis was markedly rescued by EGCG (Figure 4A). Moreover, the Bcl-2 family members are also of importance in oxidative stress-mediated neuronal death, especially the ratio of Bcl-2 to Bax [7,27]. It is the crucial marker of differentiating anti- or pro-apoptotic effects. Thus, the effects of EGCG on Bcl-2 and Bax were also investigated. The results showed that, after being treated with MG for 48h, the expression of Bax increased and that of Bcl-2 decreased. Moreover, the quantitative analysis implied that EGCG increased the ratio of Bcl-2 to Bax in a dose-dependent manner (Figure 4B,C). This was a strong indication that the decrease in the ratio of Bax and Bcl-2 caused by EGCG contributed to the prevention of MG-induced cell death.

Oxidative stress, caused by hyperglycemia and accelerating neuronal apoptosis, is also a major event in neurodegenerative disorders, which is usually a noticeable cause of diabetic complications [28]. Quercetin has been reported to reduce oxidative stress levels, activate SIRT1, and inhibit ER pathways from potentiating cognitive dysfunction [29]. Morin exhibited neuroprotective effects via the TrkB/Akt pathway against diabetes-mediated oxidative stress and apoptosis in neuronal cells. Therefore, an intervention addressing oxidative stress and neuronal apoptosis [30] is considered an essential strategy to prevent diabetic encephalopathy.

## 3. Materials and Methods

### 3.1. Reagents and Antibodies

All flavonoids (≥98% Purity) were purchased from Sichuan Weikeqi Biological Technology (Chengdu, China), and were dissolved at a concentration of 100 mM in DMSO as a stock solution (stored at −20 ℃). The solution was then further diluted in cell culture medium to create working concentrations. Methylglyoxal was obtained from Aladdin. Antibodies against Bcl-2, Bax, HO-1, β-actin, and Lamin B1 were purchased from Proteintech (Wuhan, China). Antibodies against Nrf2 and Keap1 were purchased from Beyotime Biotechnology (Beijing, China). The secondary anti-rabbit or anti-mouse HRP-conjugated antibodies were purchased from Proteintech (Wuhan, China).

### 3.2. Cell Culture and Treatment

Rat pheochromocytoma PC-12 cells were provided by Chinese Academy of Sciences Cell Bank (Shanghai, China) and cultured in RPMI 1640 supplemented with 10% fetal bovine serum (FBS), 100 g/mL streptomycin, and 100 IU/mL penicillin in a humidified 5% CO_2_ atmosphere at 37 °C. Cells were co-cultured with 0.5 mM MG and flavonoids for 48 h.

### 3.3. Analysis of Cell Viability

The proliferative activity of flavonoids on PC-12 cells was determined by 3-(4, 5-Dimethyl-2-thiazolyl)-2, 5-diphenyl-2H-tetrazolium bromide (MTT) assay. Cells were cultured in 96-well plates with 1 × 10^4^ cells per well for overnight incubation and were treated with flavonoids co-cultured with methylglyoxal for 48 h. Then, the old medium was removed and the residual medium with drugs was washed away using sodium phosphate buffer (PBS). Afterwards, 100 μL of MTT solution diluted in fresh medium was added per well and the cells were incubated at 37 °C for another 4 h. After removal of the medium, 100 μL DMSO was added per well to dissolve the formazan crystals. When fully dissolved, the absorbance at 570 nm was measured by a microplate reader (Thermo Fisher Scientific, Waltham, MA, USA) and absorbance at 630 nm as reference.

### 3.4. Western Blot Analysis

Cells were lysed in RIPA buffer with protease inhibitor cocktail (MedChemExpress) after the treatment with MG and EGCG. Protein concentration was determined by BCA protein assay kit (Solarbio, Beijing, China). Same amounts of proteins were separated by SDS-polyacrylamide gel electrophoresis and transferred onto PVDF membranes (Merck Millipore, Darmstadt, Germany). The membranes were blocked with 5% skim milk for 2 h, followed by overnight incubation at 4 °C with primary antibodies Nrf2, Keap1, HO1, Bcl-2, Bax, β-actin, and Lamin B1. The membranes were washed with TBST three times, and then incubated with secondary antibodies (1:2000 dilution) for 1 h at room temperature. Finally, the bands were visualized using an ECL kit (Solarbio, China).

### 3.5. Molecular Docking

Molecular docking was performed to predict the binding sites and efficacy between Keap1–Nrf2 complex and EGCG using AutodockTools1.5.6. The crystal structure of Keap1–Nrf2 complex (PDB: 2FLU) was downloaded from the Protein Data Bank (www.rcsb.org, accessed on 16 October 2022). Before docking, water molecules were removed from protein file 2FLU. A grid box size of 166 × 140 × 134 points with a spacing of 0.431 Å between grid points was generated to cover almost the entire favorable protein binding site. The X, Y, and Z centers were 16.401, 16.672, and 7.238, respectively. In addition, the settings are as follows: maximum energy evaluation number = 25,000,000; number of generations = 27,000; mutation rate = 0.02.

### 3.6. Flow Cytometry

Cell apoptosis was detected by flow cytometry (BD Accuri C6, USA), wherein PC-12 cells were placed in 60mm plates and stained with PI and Annexin V-fluorescein isothiocyanate (FITC) according to manufacturer’s protocols (Elabscience Annexin V-FITC/PI Apoptosis Kit). The apoptotic PC-12 cells were analyzed with FlowJo_V10.8.1.

### 3.7. Statistical Analysis

All experiments were performed three times. Data are presented as mean ± standard deviation (SD). SPSS 22.0 software was used for statistical analysis, and the significant difference was determined by one-way analysis of variance (ANOVA). *p* < 0.05 was considered statistically significant.

## 4. Conclusions

In conclusion, 25 flavonoids were evaluated for their protective activity of MG-induced PC-12 cells, and the SARs analyses obtained from the assay and molecular docking data indicated that gallate and pyrogallol groups, the configuration of the 3 position in flavan-3-ol, and some positions of the hydroxyl group’s substitution in flavonol were crucial for their activities. The mechanistic study demonstrated that EGCG, the most active compound among the test flavonoids, showed neuroprotective effects that were mediated by antioxidant and anti-apoptotic activities induced via the Nrf2/Keap1/HO-1 and Bcl-2/Bax pathways. Our studies may provide a method for rapidly screening neuroprotective antioxidants, which would contribute to the development of diabetic encephalopathy treatments.

## Figures and Tables

**Figure 1 molecules-27-07804-f001:**
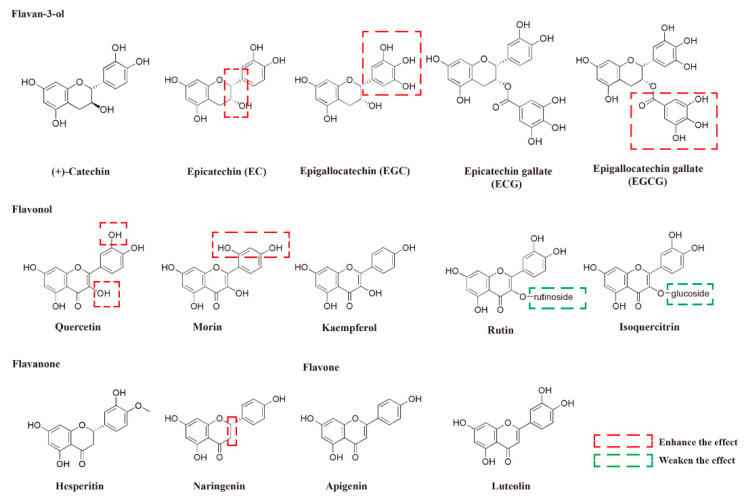
The main flavonoids tested in protective activities against MG-induced PC-12 cells.

**Figure 2 molecules-27-07804-f002:**
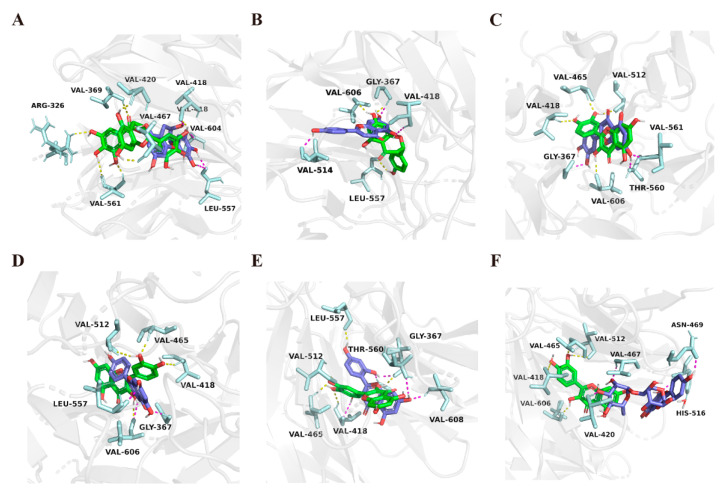
EGCG ((**A**) in green), EGC ((**A**) in purple), Naringenin ((**B**) in green), Apigenin ((**B**) in purple), Quercetin ((**C**–**F**) in green), Luteolin ((**C**) in purple), Kaempferol ((**D**) in purple), Morin ((**E**) in purple), and Rutin ((**F**) in purple) in the binding site of Keap1–Nrf2 complex. (PDB ID: 2FLU). The figure was generated using PyMol (http://pymol.sourceforge.net/) (accessed on 5 July 2022).

**Figure 3 molecules-27-07804-f003:**
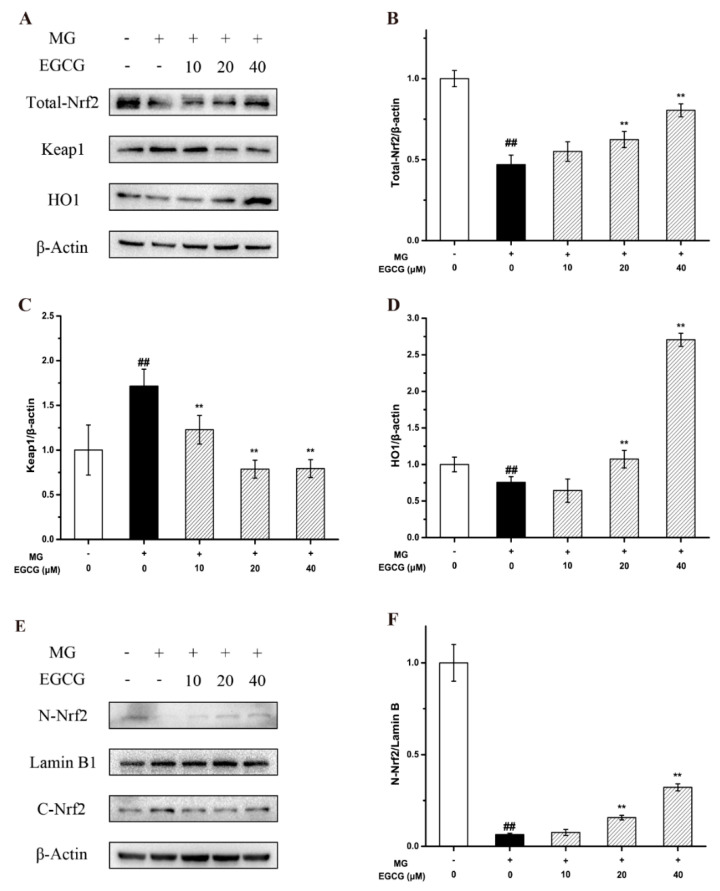
Effects of EGCG on MG-induced PC-12 cells of the Nrf2/Keap1/HO-1 pathway. (**A**–**D**) Western blot results and statistical analysis of the expression levels of Nrf2, Keap1, and HO1. (**E**,**F**) Western blot results and statistical analysis of the expression levels of nuclear Nrf2. Data shown are the mean ± SD of three independent experiments. ## *p* < 0.01 compared with the control. ** *p* < 0.01 compared with the MG group.

**Figure 4 molecules-27-07804-f004:**
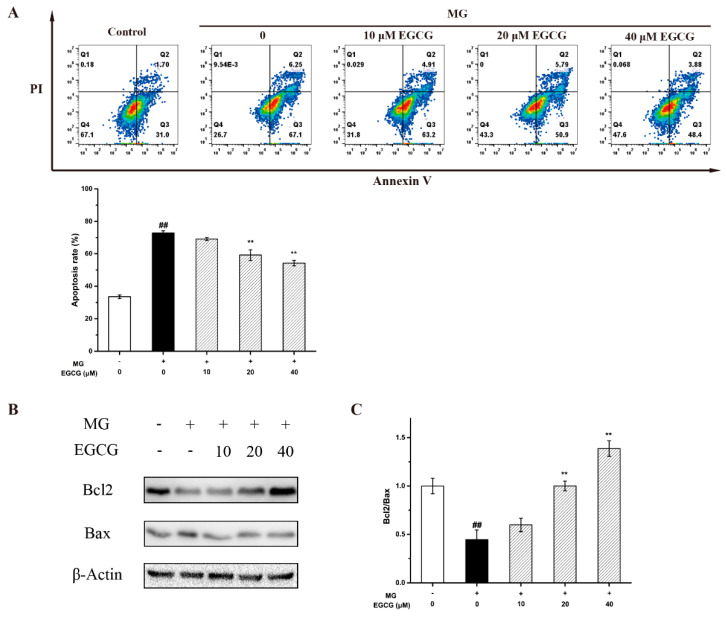
Effects of EGCG on MG-induced PC-12 cells’ apoptosis. (**A**) Analysis of cell apoptosis by flow cytometry. (**B**) Western blot results of the expression levels of Bcl2 and Bax. (**C**) Statistical analysis of the expression levels of the ratio of Bcl2 and Bax. Data shown are the mean ± SD of three independent experiments. ## *p* < 0.01 compared with the control.** *p* < 0.01 compared with the MG group.

**Table 1 molecules-27-07804-t001:** The protection activities and molecular-docking data of main flavonoids.

	Compound	EC_50_ (μM)	Binding Energy (kJ/mol)	Number of H-Bonds
Flavan-3-ol				
	(+)-Catechin	>100	−24.811	3
	EC	52.34 ± 3.99	−35.313	3
	EGC	43.06 ± 1.18	−35.439	4
	ECG	34.52 ± 2.69	−39.790	5
	EGCG	11.98 ± 0.49	−40.041	8
Flavanone				
	Naringenin	13.35 ± 1.92	−35.271	3
	Hesperitin	48.14 ± 2.31	−34.058	3
Flavonol				
	Kaempferol	>100	−32.719	3
	Quercetin	28.83 ± 2.76	−34.351	4
	Morin	14.83 ± 1.70	−34.895	6
	Isoquercitrin	>100	−22.175	3
	Rutin	>100	−29.079	4
Flavone				
	Luteolin	>100	−33.932	3
	Apigenin	>100	−33.639	3

The values shown are the means ± standard deviation of triplicate assays.

## Data Availability

Not applicable.

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
