# Peer review of "Protective Effect of Flavonoids against Methylglyoxal-Induced Oxidative Stress in PC-12 Neuroblastoma Cells and Its Structure–Activity Relationships"

_molecules, 2022, doi:10.3390/molecules27227804_

Round 1

Reviewer 1 Report

Summary

This paper mainly analyzed the structure-activity relationship of 25 flavonoids and their protective effect by in vitro methylglyoxal-induced oxidative stress model in PC12 and Keap1-Nrf2 complex molecular docking method and found several important positions and groups. The results of molecular docking fit well with the in vitro assay which indicate molecular docking might be the alternative methods for neuroprotective assay. What’s more, the Nrf2/Keap1/HO-1 pathway was further analyzed after EGCG treatment by WB and the results provided evidence for the molecular docking.

This study is of interest to the field as molecular docking might be an excellent method for peer-screen due to its strength to lower the cost and increase efficiency greatly. However, this reviewer feels the amounts of flavonoids is not meet the needs, especially just 2 flavanones and both of them show relatively strong protective activity.

Comments:

Line 37&46 Add space before the sentence begins.

Line 89-93 The results detected by MTT of MG and curcumin could provide in supplementary, thus might be more convincing than just show IC50 or EC50 values.

Line 98 remove the first “were”.

Line 130 Please add the full name for the abbreviation “CAA” first appears.

Line 192-194 It might be greater for readers to understand if some flavonoids could be exhibited in one figure in different colors when be compared, for example, Line 173-176 compared the naringenin and apigenin and explained the potential mechanism.

Line 198&199 use “in vitro”.

Line 221-223 reference 7 couldn’t support what authors want to say, because this study didn’t mention the relationship between Bcl2 family and oxidative stress-mediated neuronal death. Please check.

Line 229-233 Figure 4 As the Bcl2 and Bax were selected to illustrate the EGCG have anti-apoptotic activity. It will be better to add the results of flow cytometry.

Line 248 remove “antibodies”.

Reviewer 2 Report

In the present paper Zhang and colleagues, study the protective effect of flavonoids against Methylglyoxal-Induced Oxidative Stress in PC-12 Neuroblastoma Cells, investigating also their interaction with keap1-Nerf2. Overall the paper is well written and designed, I will just suggest the authors describe in more detail the material and method section. In particular, I will appreciate a more accurate explanation of how they performed their docking prediction (i.e. temperature, docking searching area, pH, etc..).
